# A Newly Digitised Ice-penetrating Radar Data Set Acquired over the Greenland Ice Sheet in 1971-1979

Nanna B. Karlsson[1], Dustin M. Schroeder[2,3], Louise Sandberg Sørensen[4], Winnie Chu[5], Jørgen Dall[4], Natalia H. Andersen[4], Reese Dobson[2], Emma J. Mackie[2,6], Simon J. Köhn[4], Jillian E. Steinmetz[2], Angelo S. Tarzona[5], Thomas O. Teisberg[3], and Niels Skou[4]

[1]Department of Glaciology and Climate, Geological Survey of Denmark and Greenland, Copenhagen, Denmark
[2]Department of Geophysics, Stanford University, Stanford, CA 94305, USA
[3]Department of Electrical Engineering, Stanford University, Stanford, CA 9430, USA
[4]DTU Space, Technical University of Denmark, Lyngby, Denmark
[5]School of Earth and Atmospheric Sciences, Georgia Institute of Technology, Atlanta, Georgia, USA
[6]Department of Geological Sciences, University of Florida, Gainesville, USA

**Correspondence:** N. B. Karlsson (nbk@geus.dk)

**Abstract.** We present an ice-penetrating radar data set acquired over the Greenland ice sheet by aircraft during the years 1971, 1972, 1974, 1978, and 1979. The data set comprises over 177,000 km of flight lines and contains a wealth of information on the state of the Greenland ice sheet including information on ice thickness and englacial properties. During data collection in the 1970s, the data were recorded on optical film rolls and in this manuscript, we document the digitization of these film rolls and their associated geographical information. Our data digitization enables interaction with and analysis of the data and facilitates comparison with modern-day radar observations. The complete data set in full resolution is available at the Stanford Digital Repository (https://doi.org/10.25740/wm135gp2721) with the associated technical reports. Part of the data set is available as low-resolution JPG files at The Technical University of Denmark's data repository with associated technical reports and digitized geographical information (https://doi.org/10.6084/m9.figshare.c.7235299.v1). The Stanford Digital Repository serves as a long-term storage, providing archival historic preservation in perpetuity and is not intended as a primary data access point. The DTU data repository serves as a primary entry point for data access with files organised according to acquisition year and flight line in a simple folder structure. Here, we release the full data sets to enable the larger community to access and interact with the data (Karlsson et al., 2023).

## 1 Introduction

The use of ice-penetrating radar (or radio-echo sounding) in glaciology has proliferated in the past decades, resulting in numerous insights into the state and dynamics of ice sheets and glaciers. Early studies of the polar ice sheets were dedicated to obtaining information on ice thickness and bed topography (e.g., Drewry et al., 1982; Bamber et al., 2003), but today, radar data are used to uncover information on a wealth of processes (cf., Schroeder et al., 2020). For example, studies have leveraged radar data to investigate subglacial conditions such as the presence of water, the amount and variability of basal melt, and switches between frozen and thawed conditions (Gudmandsen et al., 1975; Dahl-Jensen et al., 2003; Oswald and Gogineni,

2011; Buchardt and Dahl-Jensen, 2007; MacGregor et al., 2016b), englacial temperatures (Matsuoka et al., 2010; MacGregor et al., 2015), past accumulation rates (Nielsen et al., 2015; Karlsson et al., 2016, 2020; Lewis et al., 2017), paleo ice-flow patterns (Rippin et al., 2003; Karlsson et al., 2009; Bingham et al., 2015; Franke et al., 2022), and the chronology of the ice column (Gudmandsen, 1975; Karlsson et al., 2013; MacGregor et al., 2016a; Parrenin et al., 2017; Winter et al., 2019; Cavitte et al., 2021; Bodart et al., 2021). By utilizing the polarization of the radar energy signal, studies have also extracted information on the crystal-orientation fabric at depth (e.g., Hargreaves, 1977; Jordan et al., 2019; Dall, 2020; Ershadi et al., 2022).

In recent decades, the Greenland ice sheet has been surveyed extensively by ice-penetrating radar. The Center for Remote Sensing of Ice Sheets (CReSIS) at the University of Kansas conducted surveys starting in 1993 (Gogineni et al., 2001), and the NASA-funded Operation IceBridge began in 2009 and ran for approximately ten years in order to bridge the gap between NASA's ICESat satellite missions (see MacGregor et al., 2021). Data generated during these field campaigns spanning the 1990s to 2019 have been used in numerous studies (see references above and in Schroeder et al. (2020)), and part of the reason for the successful application of the data is the the ease of access to the data which in turn enabled large-scale analyses of ice properties.

Prior to the "modern" campaigns of the 1990s and 2000s, multiple surveys were conducted in the 1970s, where the radar data were recorded on optical film (see Fig. 1). The derived ice thicknesses have subsequently been digitised and included in compilations of ice thickness (Bogorodsky et al., 1985; Letréguilly et al., 1991; Bamber et al., 2001) although the state-of-the-art ice thickness maps from BedMachine (Morlighem et al., 2017) do not include the 1970s radar data presented here likely due to the positional uncertainties (see below). The radargrams ("Z-scopes", see an example in Fig. 4) were not digitised nor subjected to any systematic or automated analysis due to their analogue format.

Here, we present the newly digitised radargrams from the 1970s radar surveys. The high quality of our newly digitised radargrams, showing clear englacial layering and basal signal, offers unique insights into the state of the Greenland ice sheet 50 years ago. For example, tracing the englacial layering will make it possible to extend the work of MacGregor et al. (2016a) into areas that have not been covered due to observational gaps in modern radar data. The digital format also lends itself to newly developed methodologies for automatic extraction of layer slope (Sime et al., 2011; Panton and Karlsson, 2015; Holschuh et al., 2017) or quantification of layer continuity (Karlsson et al., 2012), with the potential to reveal changes in ice-flow structure or folding stratigraphy. Importantly, the data may also highlight changes in the current state of the Greenland ice sheet compared to 50 years ago. As satellite observations were not prevalent in the 1970s, the radargrams offer a multidecadal time perspective that is especially suited for studying the fast-changing margins of the polar ice sheets. For example, the Antarctic counterpart to our dataset revealed a thinning of between 10 and 33% of an ice shelf in West Antarctica between 1978 and 2009 (Schroeder et al., 2019).

## 2  Data set description

Field campaigns were carried out in 1971, 1972, 1974, 1978, and 1979 (typically in the spring) with some test flights in the late 1960s. All campaigns made use of aircraft logistics from the Antarctic Development Squadron, VXE 6, operated by the United

| Acquisition year | Flight lines acquired | Geographical coordinates digitised | Flight lines scanned |
|---|---|---|---|
| 1971 | 1, 2 | All | Unknown |
| 1972 | 1, 4, 5, 6, 7, 8, 10, 11 | All | Unknown |
| 1974 | 3, 4, 5, 6, 7, 8, 9, 10, 11, 12 | 5, 6, 8, 10, 11 | All |
| 1975 | Unknown | No | 6 |
| 1978 | 1, 2, 3, 4, 5, 6, 7, 8, 10, 11, 28 | All | All |
| 1979 | 1, 2, 3, 4, 5, 6, 7, 8, 10, 11, 18 | All | All |

**Table 1.** Overview of flight campaigns and flight numbers including information on which flight lines have digitised geographical information available. The first column indicates the acquisition year of the flight campaign, and the column "Flight lines acquired" lists the flight line numbers of that year according to the technical reports. To the best of our knowledge, we have scanned all flight lines from all years, however, the lack of metadata makes it difficult to verify if some sections are missing particularly for the years 1971 and 1972 where the film rolls were not imprinted with year and flight number. This is why they are labelled "Unknown". Please refer to Table 2 for a list of data that have been catalogued, i.e., where scanned files have been assigned acquisition year and flight line number.

States Navy. The 1960's test flights are not well documented and we have not been able to uncover data records from them. Parallel with the efforts to map the Greenland ice sheet, the Antarctic ice sheet was mapped in a joint venture between the Scott Polar Research Institute, University of Cambridge, UK, the National Science Foundation, USA, and DTU during 1971–1979. The Antarctica data set and its properties have been described in Drewry et al. (1982) and Bingham and Siegert (2007), and was recently digitised (Schroeder et al., 2019). More information on the data types and the radar system can also be found in these manuscripts and references therein.

In the following section, we briefly outline the original data set and describe our efforts to scan and digitise data and coordinates (Fig. 3).

## 2.1 Original data set

The data were collected by the then Electromagnetics Institute at the Technical University of Denmark (DTU, now part of the DTU Space department), in a collaboration between Danish, Swiss and American scientists within the Greenland Ice Sheet Program (Overgaard, 1984). The radar system used two different centre frequencies, 60 MHz and 300 MHz (Gudmandsen et al., 1975). The output signal from the sounder was presented on an oscilloscope screen as an intensity-modulated sweep which was then projected onto a 35 mm film through a camera lens (Overgaard, 1984). A schematic of this setup is shown in Fig. 1. The film speed was 1/15th of an inch per second. Radar traces were processed onboard and the films were later compressed horizontally and enlarged vertically by 10 and 6 times, respectively. This is the data presented here: images of the electronic signals in the form of radargrams. Thus, due to the way that the data were stored we do not have individual traces except in a few cases where A-scopes for selected flights were recorded but we exclude those from our dataset due to lack of metadata. No "raw" data records exist in an analogue or digital format. The radargrams have the native vertical extent of two-way traveltime and a few of our radargrams have embedded vertical markings – also called a pulse train – indicating points of

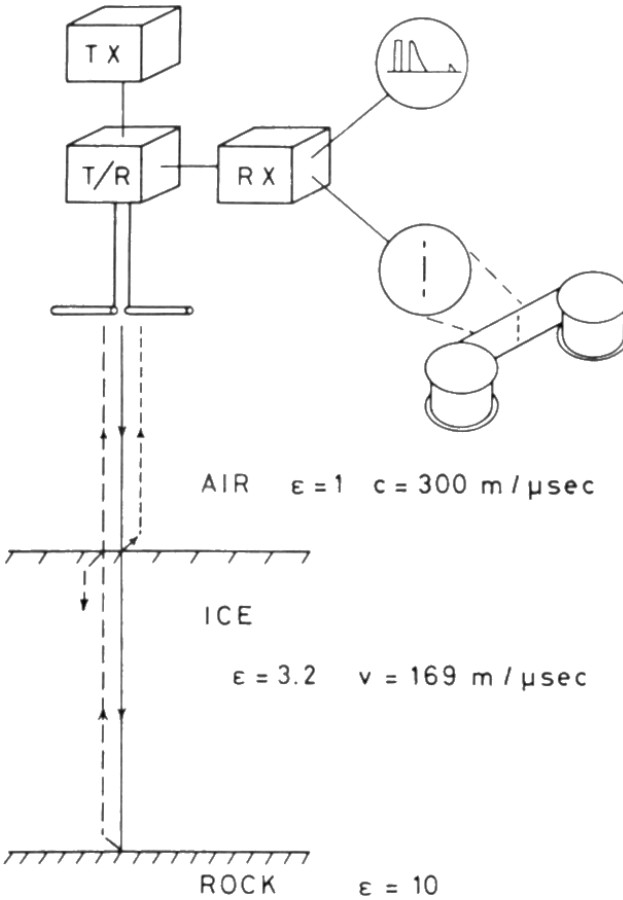

**Figure 1.** Schematic of the radio-echo sounding principle from Overgaard (1984). Tx and Rx denote transmitter and receiver. They share the antenna through the transmit/receive switch, T/R. The electrical conductivity $\varepsilon$ is noted for each media that the radar signal travels through with corresponding signal velocity.

equal time of return (for an early example of this type of marking see Figure 1 in Overgaard and Gundestrup (1985)). The vast
majority of our data does not have this vertical marking and thus the vertical extent can only be obtained from the geographical information tied to the radargram, namely, the ice thickness as recorded in the coordinate books (see Section 2.2.2 below). An example of how to obtain this information is given in Section 4.

The position of the aircraft was calculated using an inertial navigation system with an accuracy of three nautical miles (5.6 km). The aircraft position was tied to the radar data through the internal Coded Binary Decimal (CBD) that was recorded on the radar
films for each nautical mile flown. Checks on the navigational system were performed when the aircraft passed recognizable landmarks (e.g., fjords and outlet glaciers), but in the interior of the ice sheet such landmarks are absent. Anecdotally, the aircrew was able to relocate features in the ice that had been observed on previous flight campaigns (P. Gudmandsen, pers. comm.) and the positional uncertainty of three nautical miles can be considered as an upper limit.

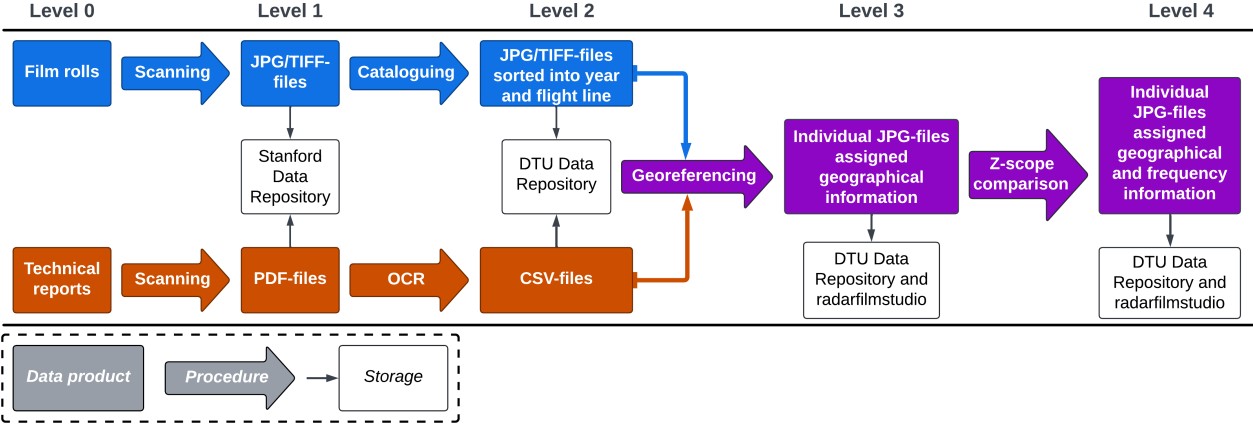

**Figure 2.** Flowchart describing the different steps the data have undergone during our digitization. Blue colours refer to the radargrams and red colours to the coordinates. Purple colours indicate merging of the two types of data. In the flowchart, OCR is short for Optical Character Recognition.

## 2.2 Data digitisation

The radar data set consists of two separate analogue archives: the film rolls as described above and their associated geographical coordinates documented in coordinate books and published as technical reports (Overgaard, 1984). The coordinate books contain geographical coordinates tied to the CBD as well as surface and bed elevation. In the following, we describe the steps we have taken during the digitization. The process is also summarised in Fig. 2.

### 2.2.1 Scanning of film rolls

The film rolls were scanned using a LaserGraphics 35-mm optical film digitization system resulting in 35-mm frames at 1,800 × 2,400 resolution with 16 bits per sample. The frames were stitched together digitally using the Python package OpenCV into 20-frame-wide radargrams. The stitched frames constitute the bulk of our data set (in terms of data volume). They are available in high-resolution TIFF format and in a lower-quality JPG format. This is part of our level 1 dataset (see Fig. 2). We note that the original films were also further compressed and made available in micro-fiche copies and they have also been scanned but 95 due to their lower resolution compared to the film rolls they have not been catalogued.

In the Results Section, we discuss how a subset of the JPG files have been catalogued, i.e., successfully tied to acquisition year and flight line number. This is part of our level 2 dataset (see Fig. 2).

### 2.2.2 Resolution and uncertainty

For the JPG images, the vertical resolution is approximately 10 m/pixels and the horizontal resolution is 2.5-3 m/pixels. The 100 corresponding resolutions for the TIFF format images are 3 m/pixels in the vertical and 0.8 m/pixel horizontally. We note that the vertical resolution may vary depending on the settings used on the flight campaign and may even change between individual

flight lines. Thus, the resolutions stated above should be considered as guidelines. We further note that while in theory the resolution should depend on the frequency of the radar signal (in our case 60 MHz or 300 MHz), this difference in resolution is obscured by the lower resolution of the images as recorded at the time. The uncertainty of the dataset is tied not only to the resolution of the radar signal but also to the uncertainty in the positioning of the flight lines. Thus a given location with a recorded ice thickness has an inherent uncertainty from the resolution of the radar signal as well as the resolution of the scanned radargram, but also an uncertainty due to the fact that the position of the aircraft was only known within a positional uncertainty of three nautical miles. As noted by Schroeder et al. (2019), if the radargram can be positioned absolutely, for example, by identifying distinct features or landmarks (e.g., fjords and outlet glaciers), then the recorded navigational information may be improved and features in the radargrams can be identified and analysed at the radargram resolution (Schroeder et al., 2019).

### 2.2.3 Digitisation of coordinates

The technical reports from the years 1971, 1972, 1978, and 1979 (Overgaard, 1984) were scanned using a standard printer scanner with enhanced contrast. The resulting files were subsequently manually enhanced ensuring that data columns were correctly aligned, that the contrast was optimal, and that imperfections in the scanned images were minimized. The files were then processed with optical-recognition software. The technical report from 1974 could not be located at DTU and a scanned version was therefore acquired from the National Snow and Ice Database in the US. We do not know which scanning settings were used for this but the quality of the scan of 1974 technical report is lower than the ones scanned at DTU. For reference, the original scans of the technical reports are available on the DTU data repository as PDF files. This is part of our level 1 dataset (see Fig. 2).

The technical report from the 1978 campaign was processed using the ABBYY FineReader PDF software while the other reports were processed with Amazon Textract. The coordinates were converted from the native format degrees, minutes and decimal of minutes to degree and decimal degrees. The result was then quality-controlled manually. This is part of our level 2 dataset (see Fig. 2) and can be found in the folder *Geographical coordinates for ice-penetrating radar surveys in Greenland 1971-1979* (doi: https://doi.org/10.11583/DTU.25804948.v1) on the DTU data repository.

### 2.3 Geo-referencing the scanned image files

The scanned images are not geo-referenced: they are simply JPG (or TIFF files) with no geographical information embedded in them. If a user is working with a film segment and wishes to find the location of the film segment, they need the acquisition year, the flight line number, and the CBD. The CBD is imprinted on the film roll, while the year and flight line correspond to the name of the folder where the segment is stored in the DTU repository. With this information, the location of the image can be retrieved from the geographical coordinate file associated with the year and flight line.

For selected flight lines (see Table 2, column: "Georeferenced"), we have attempted to rectify this lack of embedded geographical information. Due to the considerable manual intervention required for the georeferencing, we have focussed on the data from 1978 and 1979 due to the good quality of the data and the fact that the year and flight line are embedded in the radargrams. We have constructed metadata files containing information on the CBD of each image as well as information on latitude, lon-

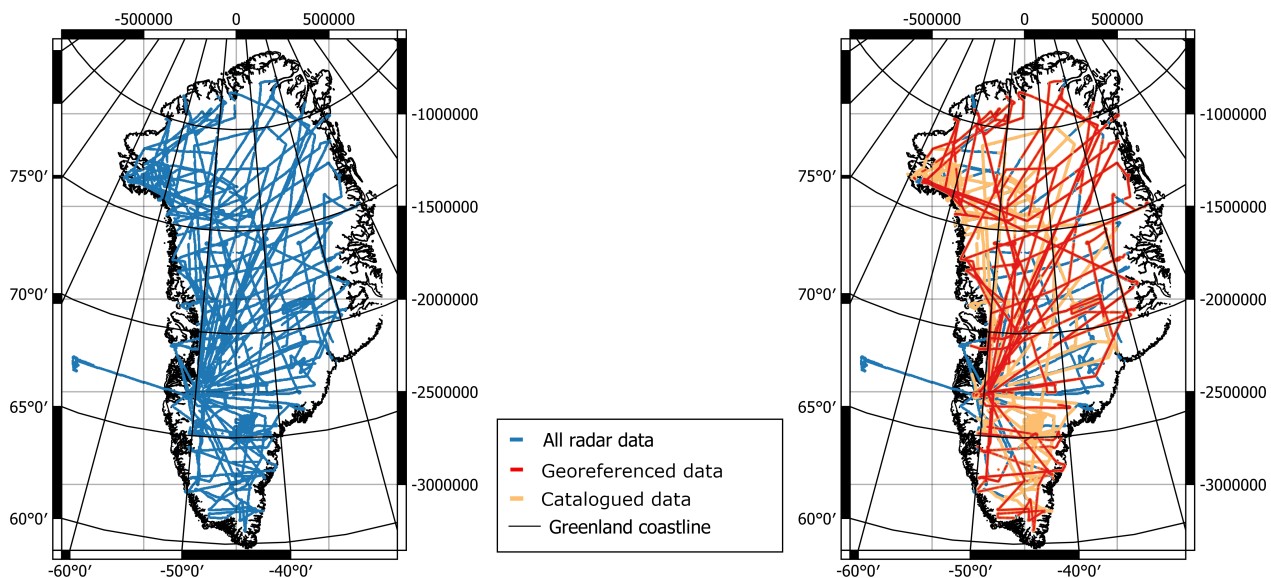

**Figure 3.** Left: Map of all radar flight lines acquired during the survey years 1971, 1972, 1974, 1978, and 1979 in blue scanned from the map in Overgaard (1984). Right: Data that have been catalogued according to acquisition year and flight number are shown in orange (level 2 data). A subset of the data that has also been georeferenced to individual image files is shown in red (level 3 data). The Greenland coastline in black is from the digital elevation model by the ESA CCI (European Space Agency Climate Change Initiative) and is based on a composite product of Cryosat-2 elevation measurements and the 5 metre resolution digital elevation model provided by the Polar Geospatial Center at University of Minnesota, USA.

| Acquisition year | Flight lines acquired | Catalogued | Georeferenced | Flight lines at DTU repository | Frequency information |
|---|---|---|---|---|---|
| 1971 | 1, 2 | N | N | N/A | N/A |
| 1972 | 1, 4, 5, 6, 7, 8, 10, 11 | N | N | N/A | N/A |
| 1974 | 3, 4, 5, 6, 7, 8, 9, 10, 11, 12 | Y | 5, 6, 8, 11 | 5, 6, 8, 11, 12 | 5, 6, 8, 11 |
| 1975* | Unknown | N | N | 6 | N/A |
| 1978 | 1, 2, 3, 4, 5, 6, 7, 8, 10, 11, 28 | Y | 1, 2, 3, 5, 6, 7 | 1-11, 18, 28, cross-pol[+] | 1, 2, 3, 5, 6, 7 |
| 1979 | 1, 2, 3, 4, 5, 6, 7, 8, 10, 11, 18 | Y | 1-11 | 1-11 | 1-11 |

**Table 2.** Overview of state and accessibility of the scanned data. The column "Flight lines acquired" is identical to the column in Table 1 and lists the flight line numbers according to the technical reports. Catalogued means that the scanned images have been tied to the acquisition year and flight line number (level 2 data). Georeferenced means that a CSV file exists for each flight line tying the filenames of scanned images to CBD and geographical information (level 3 data). For some flight lines we also have frequency information (level 4 data). *We have a scan of a flight line marked as flight 006 1975 but no technical report mentions this flight and we therefore do not have geographical information for this data. [+]Geographical information for the cross-polarized test flight has not been found but will be added to the database if they are uncovered.

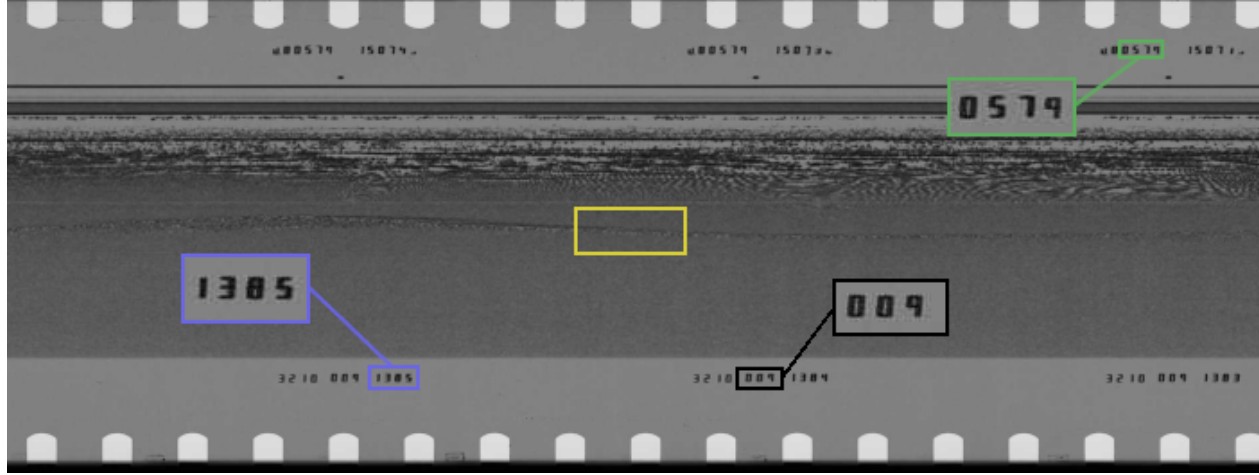

**Figure 4.** Example of a radargram. The green box indicates the date of acquisition (month and year) in this case May 1979, the black box shows the flight line number (flight line 9), and the blue box indicates the location of the CBD. The yellow box shows the location of the bed reflection. This radargram spans CBDs 1383 to 1385 corresponding to approximately 5.6 km.

gitude, and frequency (Fig. 3, red lines). This information can be found in files named *files_19YY_XX_CBD_lat_lon_Z.csv*, where XX is the flight line number, YY is the year of acquisition and Z is the frequency. Note that the images typically span several CBDs and most images are therefore listed several times for each unique CBD that they contain. The files are in a folder named *Files tying CBD and radargram filenames for ice-penetrating radar data from Greenland, 1971-1979* (doi: https://doi.org/10.11583/DTU.25295827.v1) in the DTU repository. The folder also contains shapefiles in a zipped folder with

the same information although the shapefiles are not complete compared to the CSV files. This is because we have instances where CBDs are not associated with geographical coordinates in the original technical reports but the CBDs can still be identified in the radargrams.

## 3 Results and discussion

The resulting database comprises approximately 5 terabytes of high-quality TIFF files and 2 gigabytes of identical but lower-

145 quality JPG files. We have carried out manual quality control and metadata checks on the low-resolution JPG files. For more than 80 % of the JPG files, we have manually catalogued and sorted the files into acquisition year and flight line number (level 2 data, Fig. 3, orange lines). A subset of the data has georeferenced individual image files (level 3 data, Fig. 3, red lines). Because film scans from data acquired before 1978 were not labelled with year or flight line number, radargrams and geographical information for pre-1978 data have been linked by checking radar data quality (e.g., the presence of a bed signal)

against information from the coordinate books (e.g., the recorded bed elevation), or by comparison with the micro-fiche images. Users should be aware that some of the scanned radar data might be mirrored horizontally or flipped vertically. An example of the radar data is shown in Fig. 4.

### 3.1 Database

The data can be accessed in three ways; a DTU Data repository containing the catalogued, low-resolution image files in JPG format, a Stanford Data Repository containing all scanned material as TIFF files in the original scan resolution, and a graphical interface that can be found at https://www.radarfilm.studio/map/greenland/.

We recommend using the DTU Data repository as a primary entry point. The files in the DTU Data repository are organised according to acquisition year and flight line in a simple folder structure. The geographical coordinates associated with each flight line are stored in a folder named *Geographical coordinates for ice-penetrating radar surveys in Greenland 1971-1979* containing coordinate files for all flight lines in CSV, shapefile, and GeoPackage formats. The coordinate files are named *YEAR_flX_final_QC*, where X is the flight line number. The files contain the following information: CBD, longitude, latitude, surface elevation (metres above sea level) and bed elevation (metres above sea level).

The Stanford Data Repository is meant to serve as a long-term storage and is not intended as a primary data access point. Rather, it provides archival historic preservation in perpetuity to preserve the data in the original scanned format which would enable future users to re-do everything we did "from scratch" without requiring physical access to the film itself. From the Stanford Data Repository, the complete high-resolution data set may be downloaded as single image files (contained in the folder named *DTU_originals*) or as stitched files where several film segments have been combined (contained in the folder *DTU_stitched*). In addition to the high-resolution image files, the repository contains notes related to the scanning efforts stored in the folder named *DTU_Dropbox*. We advise caution when using the data set in the Stanford Data Repository as the data have not been catalogued, i.e., assigned acquisition year and flight line. Even so, we make all the data available in the hope that any future users who undertake further cataloguing of the data will report information back to the database to enhance the value of the data and improve its usability.

One final entry point is provided for users wishing to contribute to the ongoing quality check and construction of metadata. This entry point is a graphical interface that can be found at https://www.radarfilm.studio/map/greenland/. Here, users can browse a subset of the data via a map interface and radar data can be selected based on location and downloaded in high or low quality for segments of 8 radargrams. Users can request a user account which will enable them to report errors or omissions in the data that can then be rectified.

### 3.2 Frequency information

An issue that cannot be completely resolved is the lack of information regarding frequency as the film rolls do not have information on centre frequency embedded. For flight lines where radargrams from both frequencies exist, a comparison between radargrams often reveals which is the high-frequency and which is the low-frequency acquisition (cf. Fig. 6). Typically, the lower frequency data have better penetration and thus a stronger bed signal compared to the higher frequency data. This is particularly evident in the interior of the ice sheet where ice thicknesses exceed 2 km. In contrast, the upper several hundred metres of the ice are better resolved in the high-frequency data where more individual layers are identifiable compared to the low-frequency data. We have succeeded in identifying frequency for some but not all flight lines with the help of micro-fiche

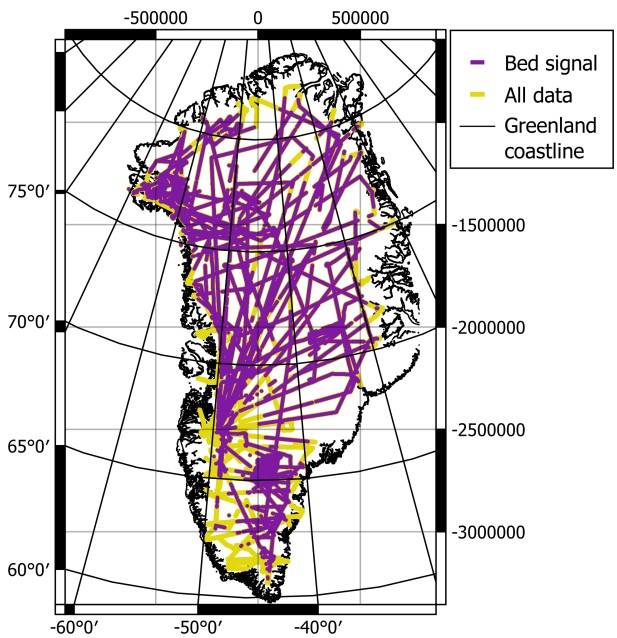

**Figure 5.** All flight lines with digitized coordinates in purple where the parts of the flight lines that have no bed signal are shown in yellow. The Greenland coastline in black is from the digital elevation model by the ESA CCI (European Space Agency Climate Change Initiative) and is based on a composite product of Cryosat-2 elevation measurements and the 5 metre resolution digital elevation model provided by the Polar Geospatial Center at University of Minnesota, USA.

images (see Table 2). However, in some cases, we only have data from one frequency and we cannot easily reconstruct the likely frequency.

### 3.3 Data quality

The quality of the data allows for easy identification of englacial and subglacial signals in many of the radargrams, such as surface and bed reflections, internal layering in the interior of the ice sheet (cf. Gudmandsen, 1975), and units of disrupted layering (cf., Panton and Karlsson, 2015). On average, 74 % of the CBD has a registered bed reflection (Fig. 5). Areas, where the signal from the bed is lost, are typically areas of warm ice (e.g., south Greenland and along the margins). Comparison with modern radar data shows many similarities (Fig. 8) and the same englacial reflections are visible in both data sets, for example, the transition between interglacial ice (lighter coloured and multi-layered) and glacial ice (darker coloured with fewer layers) which in the interior part of Northern Greenland is located one-third down in the ice column (Karlsson et al., 2013). Known data quality issues include variations in brightness, striping, lack of internal layering, and errors that were introduced when the films were developed. Fig. 9 illustrates several examples of these issues: In the top image, the bed reflection is visible but there is limited information on the internal reflections and the overall brightness of the radargram varies. On this basis, we advise caution when interpreting the absolute values of the bed reflection strength as a direct measure of bed conditions. Future work

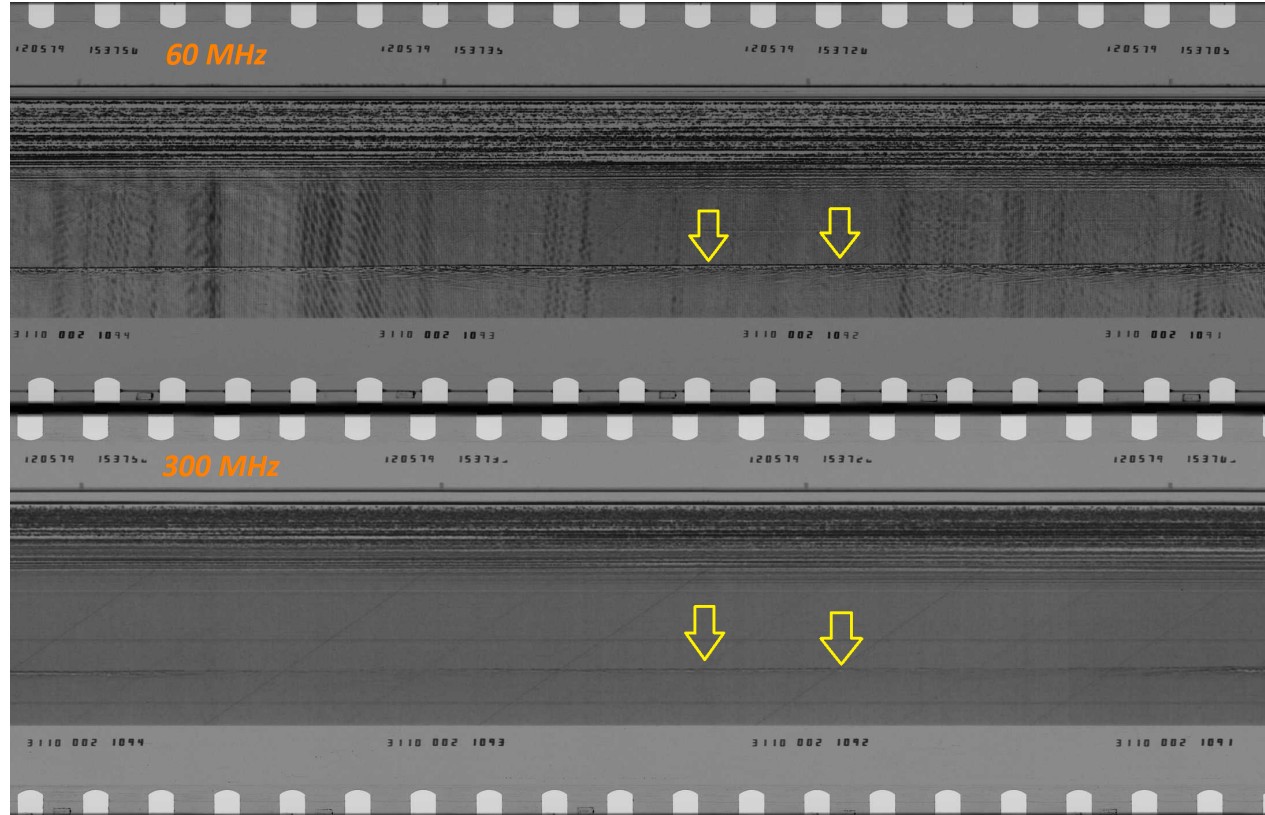

**Figure 6.** Comparison between 60 MHz and 300 MHz data from flight line 2, 1979. The yellow arrows indicate the bed reflection.

could follow the methodology of Schroeder et al. (2022) who demonstrated the use of Z-scope signals to extract radiometric observations of the subglacial environment. However, Schroeder et al. (2022) were able to leverage A-scopes for information on signal properties. Some A-scopes have been recovered from our dataset but they are currently undocumented. In the middle image, stripes obscure parts of the radargram but the bed signal is generally visible and may be interpolated across the stripes. The striping could be related to signal noise or be due to specular reflection from meltwater on the ice surface, saturating the

receiver. In the bottom image, an otherwise good-quality radargram is completely obscured by what is likely a mistake during the development of the film. It is worth noting that such issues can also be present in modern radar datasets.

## 4 Using the database

A user who is interested in only a few images for a specific location may find it more convenient to first check the https://www.radarfilm.studio/ website. For information on using the graphical interface please see the tutorial: https://www.radarfilm.

studio/docs/start. If no data are available on the graphical interface, the user may proceed to the DTU database. In the following, we outline an example of how to use the data stored in the DTU database.

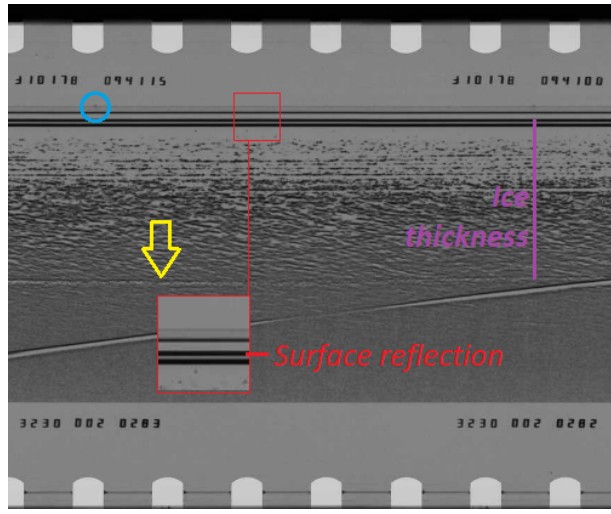

**Figure 7.** Image file 13_0014525_0014550_lowqual.jpg from flight line 2, 1978, 60 MHz. The yellow arrow indicates the bed reflection. The insert outlined in red is an enlargement of the surface reflections. The blue circle indicates the exact location of the coordinates tied to CBD 283.

A user wishing to examine the englacial properties in the area around Dye-3 Station (65°11' N, 43°49' W) during the 1970s, may proceed as follows:

 – Download the coordinates contained in the folder "Geographical_coordinates" (corresponding to the third column in Table 1).

 – Load the CSV-files, shapefiles or the GeoPackage file into a mapping software.

 – Zoom to the location of Dye-3 Station.

 – By clicking on the loaded geographical data points on the map, the user can now see that the following flight lines have been acquired in proximity to the station:

   – flight line 1, 1971

   – flight line 5, 1972

   – flight line 2, 1978

The user has now established that data exists in their area of interest. They can then proceed to the next step of identifying specific radargrams.

 – Download the coordinates contained in the folder files_CBD_filenames_info.

 – Load the CSV-files or shapefiles into a mapping software.

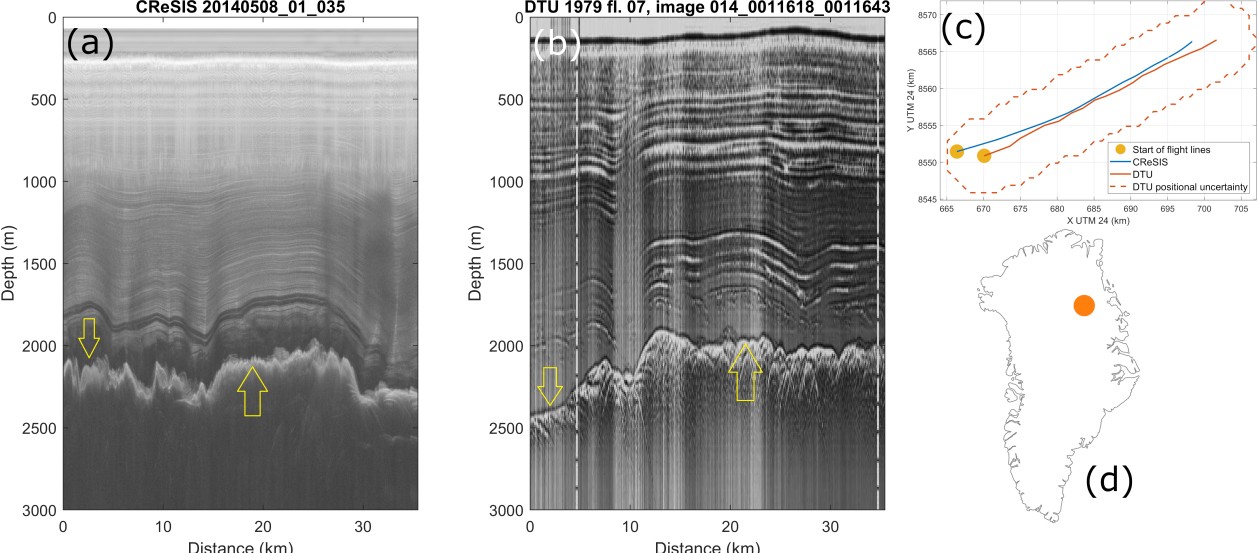

**Figure 8.** Comparison between modern-day radar data and the recently digitized DTU data. Yellow arrows indicate the basal signal. (a) shows an excerpt from a flight line acquired in 2014 by CReSIS (2023, Radar Depth Sounder Data, Lawrence, Kansas, USA. Digital Media. http://data.cresis.ku.edu/) using a frequency range of 160-230 MHz. (b) shows a 60 MHz radargram from flight line 7, 1979, that has been scaled to match the extent and horizontal compression of the modern data. The depths of both radargrams have been calculated assuming a relative permittivity of ice of 3.15. A map of the flightlines is shown in (c) with the positional uncertainty for the DTU data indicated with a dashed line. (d) is a context map with the location of (c) as a circle.

- Zoom to the location of Dye-3 Station.

- By clicking on the loaded geographical data points on the map, the user can now see that the following flight lines have been georeferenced to individual CBDs (corresponding to the third column in Table 2):

– flight line 2, 1978 (closest CBD: 282 in image 013_0014525_0014550_lowqual.jpg).

The user may now simply go to the folders of the respective years and flight lines (in this case the folder 1978_fl02) and download them. If our user is interested in knowing the exact location and ice thickness of CBD 282, then they must examine the image named 013_0014525_0014550_lowqual.jpg at CBD 282. An example of this image is shown in Fig. 7. The exact location of CBD is given by the small indicator below the time of acquisition in the upper right part of the radargram. The 235 corresponding indicator for CBD 283 is marked with a blue circle. According to the coordinate files downloaded from the Geographical_coordinates folder, CBD 282 corresponds to the location 65.167N and -43.8, surface height above sea level is 2487 m and bed height above sea level is 251 m. The ice thickness is thus 2236 m. Thus, the vertical extent of the radargram between surface reflection and bed reflection at CBD 282 is 2236 m (indicated with a purple line in Fig. 7). The horizontal extent is equivalent to the distance between the indicators which is one nautical mile.

## 5 Conclusions

We present a newly digitized radar data set acquired in the 1970s over the Greenland ice sheet. The data are now for the first time digitally accessible and may be analysed and compared against more recent radar data.

The data can be accessed as image files in high and low resolution via data repositories at Stanford University and DTU, respectively. A data portal with an interactive map is also available for community members wishing to enhance the data by reporting further metadata and carrying out quality checks. Geographical coordinates for the flight lines are available as CSV files for 75% of the data set. Scans of parts of the data set have been assigned geographical information on an image-by-image basis and this information is also available as CSV files.

While uncertainties in positioning, missing information on centre frequency and errors associated with the original film development entail a less good quality of data compared to modern-day radar data, the data set presented here nevertheless provides a unique window into the past state of the ice sheet. Given the extent and overall good quality of the radargrams as well as the existence of features that may be easily compared to features in modern-day data, the data make it possible to extract information on 5 decades of change of the state of the ice sheet.

## 6 Data availability

The entire collection of the scanned radar data is available at the Stanford Digital Repository (https://doi.org/10.25740/wm135gp2721), Karlsson et al. (2023)). The catalogued data are available at low resolution at DTU data with associated geographical information (DOI pending, temporary access at: https://e.pcloud.link/publink/show?code=kZKlYsZk5aCdD2blOyKu49wMSyR8HDVJhGV). This includes the geographical and glaciological information on latitude, longitude, ice thickness, and surface elevation as registered to the Coded Binary Decimal (CBD) counter. A graphical interface designed to enable quality checks of the scans and positioning data is available at https://www.radarfilm.studio/map/greenland/.

*Author contributions.* NBK conceived of the study together with LSS. JD and NS provided technical assistance with interpretation and data curation. NBK, LSS and JD secured funding for the purchase of the manual ScanPro 3000 Micro film scanner and NHA carried out the scans at DTU Space in 2015 and 2016. DMS secured funding for and provided the LaserGraphics 35-mm optical film digitization system. DMS, WC and EM scanned the data in the summer and autumn of 2019. NBK, WC, AST and DMS digitized the coordinates. The radarfilm.studio website was constructed by TOT. Subsequent data cataloguing and construction of metadata were carried out by NBK aided by TOT, AST, RD, and JS. SJK wrote part of the code that generated Fig.8. NS participated in data acquisition in the 1970s. NBK wrote the manuscript with input from all authors.

*Competing interests.* The contact author has declared that none of the authors has any competing interests.

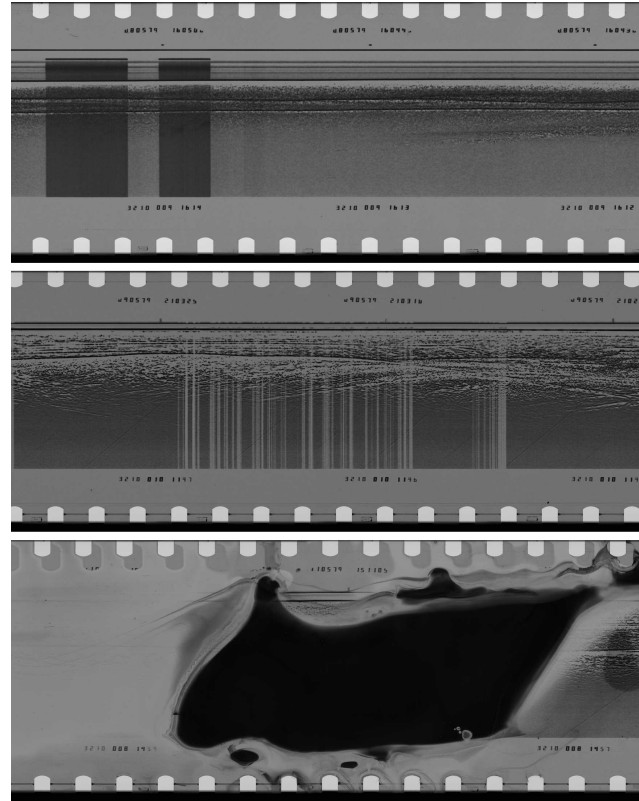

**Figure 9.** Example of radargrams acquired in 1979 from flight lines 9, 10, and 8, respectively. The radargrams exemplify different known data quality issues, such as, variations in brightness along a flight line, the occurrence of stripes, and errors introduced during the development of the films. The radargrams all span approximately 5.6 km.

*Acknowledgements.* We thank the editor, N. Dematteis, and referees J. MacGregor and J. Bodart for helpful comments and suggestions that substantially improved the manuscript.

The development of the radar was funded by National Science Foundation in collaboration with DTU. We thank Brødrene Hartmanns Fond for supporting the purchase of a ScanPro 3000 Microfilm scanner that made it possible to carry out an initial assessment of the data quality. We gratefully acknowledge Famillien Hede Nielsens Fond and P. A. Fisker's Fond who supported the 2019 scanning campaign and the associated travel. We gratefully acknowledge the support from the Carlsberg Foundation (grant CF19-0698) for a visit by NBK to Stanford University to carry out the final data assessments. DMS and EJM were supported, in part, by an NSF CAREER Award. DMS was also supported by 275 a grant from the Stanford Woods Institute for the Environment and a grant from the Heising Simons Foundation. WC was supported, in part, by a grant from the NASA Cryospheric Sciences Program and from NSF Antarctic Research (#8071686). TOT was supported in part by a NASA FINESST award (grant 80NSSC23K0271), the TomKat Center for Sustainable Energy, and Stanford Data Science. RD and JS were supported by funding from the Stanford SESUR and SURGE programs, respectively. We thank Niels Skou and Finn Søndergaard who acquired the radar data and the many engineers, technicians, and field workers who were involved in the collection of the original data sets.

Finally, we are indebted to the late Professor Preben Gudmandsen who initiated the radar ice-sounding activities at the Technical University of Denmark and whose support in digitizing his pioneering work was invaluable.

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
