# Peer review of "A Newly Digitised Ice-penetrating Radar Data Set Acquired over the Greenland Ice Sheet in 1971-1979"

_Earth System Science Data, 2023_

## Author Response (AR1)

Dear Editor Niccolò Dematteis,

Please find below our responses to the comments from the two referees.

We note that we have now obtained storage and DOI at the DTU data repository solution (FigShare). The main change compared to the temporary storage is that we were not able to upload files individually and therefore the data are now in zipped folders where each folder contains images from a single flight line from one year. This means that users cannot download individual images but must download folders. We have updated the text to reflect this. In addition, some folders have changed names to be more descriptive of what they contain. This is also updated in the manuscript text.

On behalf of all authors,

Nanna B. Karlsson

Referee #1: J. MacGregor

I am perhaps overly predisposed to appreciation of studies such as this one, which seek to resurrect long-ago datasets into a modern form so that they may provide insight into long-term ice-sheet behavior. Nevertheless, this MS is an excellent example thereof. The data are lucidly described, and the process of sorting out the wheat from the chaff in the various campaigns is rigorous. The examples provided of the radargrams are clear, and the suggested process for accessing the data, which I reproduced, is good. I have a couple larger concerns/perspectives that may or may not be resolvable within the context of an ESSD submission, and some other minor comments.

*We thank the referee, J. MacGregor, for their support and enthusiasm for our work. We address the individual comments below.*

Larger interrelated concerns:

1. Conspicuously missing from the list of ice thickness compilations on L32 for which these 1970's DTU data have been used is Morlighem et al. (2017, GRL), i.e., BedMachine v3. This is the only ice thickness compilation in modern use, is regularly updated (now v5) and is the de facto standard for Greenland. In the past, it has not been clear to me whether these 1970's data were included in BedMachine. The legend of Figure 1a in Morlighem et al. (2017) contained just enough ambiguity to make that unclear. However, the shapefiles provided at the DTU repository provided an unambiguous answer to this question when compared against the dataid field of BedMachine v5: they are not. While the authors fairly discuss many potential applications of the dataset, the most conservative assessment of their near-term value is ice thickness measurements in otherwise unsurveyed regions. I assume that the 3 nautical miles uncertainty, equivalent to 5.6 km or ~37 BedMachine 150-m grid cells, is the primary reason why, but I don't know for sure. A direct assessment and discussion of this issue seems essential.

*To our knowledge, it is correct that the DTU 1970s radar data presented here are not included in current versions of BedMachine. We do not know the reason for this but as the referee notes it is probably due to*

*the large spatial uncertainty of the dataset. It is our hope that the publication of the data and the added metadata may pave the way for including the dataset in later versions of BedMachine.*

*In the revised manuscript, we have added the following sentence in line 33:*

*"Prior to the ``modern'' campaigns of the 1990s and 2000s, multiple surveys were conducted in the 1970s, where the radar data were recorded on optical film (see Fig 1). The derived ice thicknesses have subsequently been digitised and included in compilations of ice thickness mappings (Bogorodsky et al., 1985; Letréguilly et al., 1991; Bamber et al., 2001) although state-of-the-art ice thickness maps from BedMachine (Morlighem et al., 2017) do not include the 1970s radar data presented here likely due to the positional uncertainties (see below)."*

2. The georeferencing synthesis ought to be highlighted sooner/better in the MS as it is fundamental to any interpretation. Further, why provide a large set of shapefiles when a substantially better and simpler one-file format is available, i.e., GeoPackage? See http://switchfromshapefile.org/ for relevant arguments. Further, it is not obvious from the filename or description in the MS that the surface and bed elevations are also included in those shapefiles, enabling calculation of the ice thickness directly. Providing an example QGIS style file for such a calculation, rather than forcing every user to generate their own QGIS style expression, would be helpful.

*GeoPackage: This is a great suggestion. We have generated a GeoPackage file that contains the same information as the shapefiles. We have added the following to lines 130-133:*

*"The geographical coordinates associated with each flight line are stored in a folder named* Geographical coordinates for ice-penetrating radar surveys in Greenland 1971-1979 *containing coordinate files for all flight lines in CSV, shapefile, and GeoPackage formats.*

*At the suggestion of referee #2, we have restructured some of the georeferencing sections and changed terminology. We hope this addresses the concerns of the referee.*

*We do not think that generating a QGIS-style file for the information is helpful for the users as QGIS is only one of numerous software packages that handle mapping. We leave this to the discretion of the editor whether it is required.*

3. If the 3 nautical mile uncertainty can indeed be considered an upper limit, are there places where the uncertainty is known to be lower and could this be added as a field to the georeferencing information?

*As we note in lines 71-72, the positional uncertainty is lower close to recognizable landmarks which means at the beginning of flight lines and close to known areas along the ice sheet margin or in the interior. We are hesitant to flag this further in our dataset as this is not immediately quantifiable. We have added the following sentence:*

*"…three nautical miles can be considered as an upper limit. For example, at the beginning of each flight line, the positional uncertainty is substantially lower as the initial position was well-known."*

Minor comments:

L32: "mappings" seems superfluous

*"Mappings" deleted.*

L44: comma missing after "signal"?

*We are unsure of this comment. There is a comma after "signal" in our version of the manuscript.*

44: "unprecedented" seems like not the right word, given that Schroeder et al. (2019, PNAS) achieved something similar for Antarctica, and other pre-digital archival records have been digitized to similar beneficial effect (e.g., Kjeldsen et al., 2016, Nature)

*In this context, unprecedented refers to the insights into the properties of the Greenland ice sheet. We have changed it to "unique" but note that neither that Schroeder et al. (2019) nor Kjeldsen et al. (2016) reveal insights into the englacial properties of Greenland.*

47: comma missing after VXE 6?

*Comma added.*

62: simplify to "of two-way traveltime"? not sure why parenthetical statement chosen here

*Good point. The sentence has been changed to:*

*"The radargrams have the native vertical extent of two-way traveltime and…"*

106: is \*lower\* than the ones scanned at DTU

*Changed. Thank you.*

110: "film originals" meant?

*Errant "s". Changed to:*

*"We note that the original films…"*

135: "from scratch" is a useful colloquialism but perhaps worth contextualizing with something like the following: "without requiring physical access to the film itself".

*Good suggestion. The suggested sentence has been added.*

182-3: This statement is fair but seems like an obtuse reference to the efforts of Schroeder et al. (2022, JGlac) on the Antarctic film record, without citing that study. Also, why would a correction for apparent

internal reflection power, similar to Gades et al. (2000, JGlac) for Siple Dome, not work in this context? IRP variations are noise to some, but not necessarily a dealbreaker?

*The Schroeder et al. (2022) study used A-scopes to calibrate the Z-scope basal reflections. We haven't been able to properly document the A-scopes in our dataset. We have modified the statement "On this basis, we advise against attempting to deduce absolute values of bed reflection strength from the current data set" to the following:*

*"On this basis, we advise caution when interpreting the absolute values of the bed reflection strength as a direct measure of bed conditions. Future work could follow the methodology of Schroeder et al. (2022) who demonstrated the use of Z-scope signals to extract radiometric observations of the subglacial environment. However, Schroeder et al. (2022) were able to leverage A-scopes for information on signal properties. Some A-scopes have been recovered from our dataset but they are currently undocumented."*

Table 1: The description of the "missing" flights is good. A minor perspective from my OIB days that may help understand what happened: It may be that the flight numbers were sometimes numbered not by science concerns but as a sequence in the broader campaign, which included flight numbers for transits between Thule AFB (now Pituffik SB) and Søndrestrøm AB (now Kangerlussuaq). It does seem that flights based from both locations, given their geometry, but I didn't check if that happened in the same year. In other words, the flight numbers may have been set by the air crew, not the scientists, once the campaign began.

*This is a very useful point and probably part of the explanation for the numbering of flight lines.*

For Tables 1 and 2, it might be helpful to add columns with percentages of total know flight lines scanned or QC'd where appropriate, especially for Table 2 so that it requires a bit less referencing to Table 1.

*We have added the column "Flight lines acquired" from Table 1 to Table 2 for easier referencing.*

Figure 6:

Label individual panels a/b/c/d

For panel c (flight line map), the uncertainty of the DTU line should be shown as background fill.

*Thanks for the suggestion. We have labelled the individual panels and outlined the positional uncertainty of the DTU flight line.*

Referee #2, J. Bodart

Summary and overall comment

In their paper, Karlsson et al. discuss the digitisation and data release that they undertook on legacy airborne radar data acquired over the Greenland Ice Sheet in the 1970s. They discuss in great detail the methods used to digitise and quality check the data in order to provide a robust dataset that can be used by scientists to observe decadal-scale changes over the Greenland Ice Sheet. This work is extremely valuable, as satellite data does not go back long enough to observe such decadal changes. In combination with the release of modern radar data, the release of legacy data as done here now enables such comparisons over multi-decadal timescales, as previously conducted over the Antarctic Ice Sheet by Schroeder et al., 2019, amongst others.

I had a really great time reading the paper and diving into the legacy reports and data that are provided alongside the manuscript. I found the paper well written and for the most part (except see general comment below regarding the methods), well structured. I also found the figures very useful and easy to interpret, even to a non-radar audience. The dataset is robust, easily accessible, and relatively well described in the paper.

As it stands, I would recommend very minor revisions to the paper to address the points highlighted below. I look forward to seeing this important paper and associated dataset published in Earth System Science Data.

*We thank the referee, J. Bodart, for their support and enthusiasm for our work. We address the individual comments below.*

General comments

The abstract mentions that data are available at both high resolution (through Stanford Repository) and low resolution (through DTU). It was confusing to me reading this why you would provide both at different locations, so I wonder whether something could be done in the abstract to explain, in a very brief way, why such difference exists. Otherwise, readers reading this for the first time may, like me, spend a good part of the paper wondering why this was done (until one reaches Section 3.1 and 3.2 where it is explained why the two formats exist).

*Good point. We have changed the abstract to reflect this:*

*The complete data set in full resolution is available at the Stanford Digital Repository (https://doi.org/10.25740/wm135gp2721) with the associated technical reports. Part of the data set is available as low-resolution JPG files at The Technical University of Denmark's data repository with associated technical reports and digitized geographical information (https://doi.org/10.6084/m9.figshare.c.7235299.v1). The Stanford Digital Repository serves as a long-term storage, providing archival historic preservation in perpetuity and is not intended as a primary data access point. The DTU data repository serves as a primary entry point for data access with files organised according to acquisition year and flight line in a simple folder structure. Here, we release the full data sets to enable the larger community to access and interact with the data (Karlsson et al., 2023)."*

I would recommend moving Section 3.2 much higher in the manuscript. This links to my previous comment regarding the confusion about the low and high resolution files, and also because until the reader reaches Sections 3.1-3.2, multiple mentions of different resolutions, files, and methods used (e.g. QC of the data) are provided without appropriate context. This made the reading a little difficult to follow at first, and I spent a bit of time having to untie some knots and connect the dots by scrolling up and down the paper several time until I could understand the methods used (e.g. in the georeferencing). See my line-by-line comments below for more clarification.

*Great suggestion. We have moved Section 3.2 higher up and it is now Section 2.3 immediately following the description of coordinate digitisation.*

*We agree that the descriptions of the different types of data products are unclear. The term "quality-controlled" does not accurately describe what type of procedure has happened. To amend this, we have taken the following actions:*

1. *Included a new figure showing the different types of data products and the overall processing flow.*
2. *Replaced the term "quality-checked" with the more appropriate term "catalogued" when referring to sorting the image files into year and flight line number.*
3. *We have introduced the terminology "levels 1-4 data" based on the flowchart describing the processing.*

*Several parts of the manuscript have been updated to reflect this change.*

============================================================================================

Line-by-line comments:

Line 5: "Our digitisation of the data" – replace by "Our data digitisation".

*Changed.*

Line 26: "and the NASA Operation IceBridge began in 2009." – I found the end of this sentence a bit abrupt. Consider adding: "began in 2009 for approximately ten years in order to bridge the gap between NASA's ICESat satellite missions (see MacGregor et al., 2021)". Reference: https://doi.org/10.1029/2020RG000712

*Changed.*

Line 28: "is the easily accessible data format" – rephrase for "is the ease of access to the data". I would argue that the words "easily accessible" can be nuanced here, primarily because even though the data is easy to find, the data format of the OIB surveys (i.e. matlab files) do not technically comply with the FAIR data standards, since MATLAB is a proprietary software.

*Changed.*

Line 28-29: "where the immediate digitisation of the data made it possible to carry out large-scale analyses of ice properties" – this sentence is a bit confusing to me. Could you rephrase as follows: "ease of access to the data, which in turn enabled large-scale analyses of ice properties."

*Changed.*

Line 33: "radargrams ("Z-scopes")". Add mention of Fig. 3 here when mentioning Z scopes for the first time here.

*Done*

Line 34: Add new paragraph between Line 33 and Line 34

*Done*

Line 43: Add Schroeder et al. 2019 reference here after "West Antarctica between 1978 and 2009."

*Done*

Lines 43-44: starting with "The high quality of our newly"- I would recommend to move this sentence to Lines 34-35 to replace the current sentence there (which starts with "The digitisation will make it possible to expand"). I think it's a much better introductory sentence which would link up well with the following sentences and I'm not sure it has much value where it currently is at the end of the paragraph.

*Changed.*

Table 1: Perhaps explain briefly in the caption why some "Flight lines scanned" are marked as "Unknown". And refer to the section which discusses this.

*We have added a sentence to the caption so it now reads:*

*"To the best of our knowledge, we have scanned all flight lines from all years, however, the lack of metadata makes it difficult to verify if some sections are missing particularly for the years 1971 and 1972 where the film rolls were not imprinted with year and flight number. This is why they are labelled "Unknown"."*

Table 1 caption: "Please refer to Table 2 for a list of" – this is the first instance since the abstract that the quality check is mentioned, and it clarifies a little bit what has been done. However, I think it's a bit too low in the manuscript and should be explained briefly when first mentioned (see general comment above).

*We hope with the change in terminology outlined above this is no longer an issue.*

Line 62: "(or rather in two-way travel time)" – replace "or rather in" with "in radar".

*This has been changed at the suggestion of referee #1.*

Lines 73-77: I found this paragraph to read a bit out of place. I think it's important to mention, but it should be placed further up in the introduction section in my opinio.

*The paragraph has been moved to the Section 2 Data set description*

Line 98: "landmarks" – can you be more specific? i.e.: "such as basal channels and bedrock features"

*We have added "(e.g., fjords and outlet glaciers)" to the sentence to align with line 70.*

Line 110: "on the data website" – which one? The Stanford repository or the DTU? Be specific

*Has been changed to "DTU data repository". Thanks!*

Lines 115-118: I'm not sure I understand the difference between data that has been qced with geographic information (80%; orange line) and data that has been qced and georeferenced (red line)? Is the difference the sentences following this (Lines 118-121)? This is not super clear to me and is worth expanding or clarifying.

*We hope with the change in terminology outlined above this is no longer an issue.*

Table 2: The caption mentions cross-polarised flights were conducted, but no mention of this is present in the text. I also see that there is a file "1978_fl111_crosspol". I have two comments on this: (1) could you provide more details on the extent of those cross-polarised flights that were conducted as part of these 1970s surveys discussed in the paper? And (2) what is meant in the caption by "Geographical information is not yet available for the cross-polarized test flight" – the word "yet" implies this is work in progress? If so, how could users access this information in the future?

*We haven't been able to find information or coordinates for the cross-polarized flights. We include them in Table 2 for completion but we do not know if the data exist. They might still be somewhere in the DTU basement. The sentence "Geographical information is not yet available for the cross-polarized test flight." has been changed to "Geographical information for the cross-polarized test flight has not been found but will be added to the database if they are uncovered."*

Line 140: what is meant by "users will report"? Maybe clarify this point.

*The sentence has been changed to:*

*"Even so, we make all the data available in the hope that any future users who undertake further cataloguing of the data will report information back to the database to enhance the value of the data and improve its usability."*

Lines 149-150: Consider reversing the order of the variables mentioned in the text so that they match the csv header order (year, flightline number, and cbd number)

*Done.*

Section 3.2: I found this section really useful, but it comes much too late in the paper. As my comments above (general and line-by-line) indicate, I was a bit confused with the structure of the text, particularly at the start when mentioning different resolutions in the files and different methods used (i.e. the qaqc which was applied only on a subset of the data). I had to switch back and forth between different sections of the paper to link them together. As a result, I think Section 3.2 should be moved before Figure 2 and Table 2, at least. This is particularly true for Lines 148-150 which should also definitely be moved before Section 2.2.1, at least.

*We hope with the changes outlined above this is no longer an issue.*

Line 153: "For selected flight…" – why not for all? Is it because of this sentence in the caption of Table 1 higher up in the paper: "particularly for the years 1971 and 1972 where the film rolls were not imprinted with year and flight number"? If so, please specify this again as this sentence can be overlooked at the start of the manuscript.

*The short answer is that it is an incredibly time-consuming process to georeference individual image files. While some progress has been made using AI, the georeferencing requires a lot of human interaction. We have added the following sentence:*

*"Due to the considerable manual intervention required for the georeferencing, we have focussed on the data from 1978 and 1979 due to the good quality of the data and the fact that the year and flight line are embedded in the radargrams."*

Line 155: Ah ok! So this is what the red lines are. This partially addresses my comment above. Again, I would recommend moving this section higher up and make sure Figure 2b follows this explanation. I would also recommend adding at the end of Line 132 that this sentence is for the orange line in Figure 2b (if I understood completely!). I also now note that the caption of Table 2 explains the difference between qced and georeferenced or qced with geographic information, but this comes after having read this section multiple times and trying to tie the knots together.

*We hope with the change in terminology outlined above this is no longer an issue.*

Line 163: "regarding frequency. The film rolls" – join both sentences together; i.e. "regarding frequency, as the film rolls"

*Done*

Line 181: Figure 8 is mentioned before Figure 7 in the text.

*The order of the figures has been changed. Thanks for spotting that.*

Line 187 and end of this paragraph: Consider adding "It is worth noting that such issues can also be present in modern radar datasets". I think it's important to note that these can be common issues that are not necessarily down to your specific dataset, or the acquisition methods used at the time.

*Good point! Thanks.*

Figure 6: Provide sub-panel numbering for each sub-figure. Link these too in the caption (i.e. "modern day radar data (a) and the recently digitised DTU data (b)").

*Done*

========================================================================================
===================

---

## Author Response (AR2)

03-06-2024

Response to reviews of " A Newly Digitised Ice-penetrating Radar Data Set Acquired over the Greenland Ice Sheet in 1971-1979" by Karlsson et al., submitted to ESSD.

Changes in the final version:

1. On line 55 (new version) We have changed "Antarctic ice sheets" to "Antarctic Ice Sheet" as requested by referee J. Bodart.
2. The caption of Figure 5 has been updated to include a reference to the source of the Greenland coastline. The caption now reads:
   *"All flight lines with digitized coordinates in purple where the parts of the flight lines that have no bed signal are shown in yellow. The Greenland coastline in black is from the digital elevation model by the ESA CCI (European Space Agency Climate Change Initiative) and is based on a composite product of Cryosat-2 elevation measurements and the 5 metre resolution digital elevation model provided by the Polar Geospatial Center at University of Minnesota, USA."*

Once again we would like to thank the editor N. Dematteis and referees J. MacGregor and J. Bodart for their constructive reviews of the our manuscript.

On behalf of all authors,

*N.B. Karlsson*

Nanna B. Karlsson